# Unexpected Huge Prevalence of Intracardiac Extension of Wilms Tumor—A Single Center Experience from a Ugandan Hospital

**DOI:** 10.3390/children9050743

**Published:** 2022-05-19

**Authors:** Massimo Mapelli, Paola Zagni, Roberto Ferrara, Valeria Calbi, Irene Mattavelli, Manuela Muratori, Jackson Kansiime, Cyprian Opira, Piergiuseppe Agostoni

**Affiliations:** 1Centro Cardiologico Monzino, IRCCs, Via Parea 4, 20138 Milan, Italy; irene.mattavelli@cardiologicomonzino.it (I.M.); manuela.muratori@ccfm.it (M.M.); piergiuseppe.agostoni@ccfm.it (P.A.); 2Department of Clinical Sciences and Community Health, Cardiovascular Section, University of Milan, 20122 Milan, Italy; 3Terapia Intensiva Neonatale, Ospedale Fatebenefratelli P.O. Macedonio Melloni, Via Macedonio Melloni 52, 20129 Milan, Italy; paola.zagni@gmail.com; 4Medical Oncology Department, Fondazione IRCCS Istituto Nazionale dei Tumori, 20133 Milan, Italy; robertoferrara86@gmail.com; 5San Raffaele Telethon Institute for Gene Therapy (SR-TIGET), IRCCS San Raffaele Scientific Institute, Via Olgettina, 60, 20132 Milan, Italy; valeria.calbi@gmail.com; 6Pediatric Immunohematology Unit and BMT Program, IRCCS San Raffaele Scientific Institute, Via Olgettina, 60, 20132 Milan, Italy; 7St. Mary’s Hospital Lacor, Gulu P.O. Box 180, Uganda; jkkansiime@gmail.com (J.K.); opira.cyprian@lacorhospital.org (C.O.)

**Keywords:** Wilms tumor, heart diseases in sub-Saharan Africa, echocardiography

## Abstract

Wilms tumor (WT) is the most common primary renal malignancy in young children. WT vascular extension to the inferior vena cava (IVC) occurs in 4–10% of cases and can reach the right atrium (RA) in 1%. Data on WT clinical presentation and outcome in developing countries are limited. The aim of the present study is to describe the prevalence of intracardiac extension in a consecutive population of WT patients observed in a large non-profit Ugandan hospital. A total of 16 patients with a histological diagnosis of 29 WT were screened in a 6-month period. Patient n°2, a 3 y/o child, presented with a 3-week history of abdominal distension, difficulty in breathing, and swelling of the lower limbs. A cardiovascular system exam showed rhythmic heart sounds, a heart rate of 110 beats per minute, and a pansystolic murmur on the tricuspid area; the abdomen was grossly distended with a palpable mass in the right flank, hepatomegaly, and splenomegaly. An abdomen ultrasound showed an intra-abdominal tumor, involving the right kidney and the liver and extended to the IVC. An ultrasound guided biopsy showed a picture consistent with WT. Cardiac echo showed a huge, mobile, cardiac mass attached to the right side of the interatrial septum, involving the tricuspid valve annulus, causing a “functional” tricuspid stenosis. The patient died of cardiogenic shock 7 days after admission. Patient n°3, a 3 y/o child, presented with analogue symptoms and the same diagnosis. The cardiac echo showed a round mass in the RA. Thirteen more patients were screened with cardiac echo, showing a normal heart picture. In our limited series, we found WT cardiac extension in three patients over 16 (19%). Cardiac echo performed routinely can lead to a better staging, prognostic, and therapeutic assessment. In our setting, the intra-cardiac extension could be more frequent than previously reported and might have prognostic implications.

## 1. Introduction

Wilms tumor (WT) represents the second most common intra-abdominal cancer of childhood and the fifth most common pediatric malignancy (6% of all pediatric cancers). It is the most common primary renal malignancy (95%) in young children [1,2,3]. Multimodal strategies have led to a dramatic improvement in these patients’ prognoses. In fact, with optimized use of current treatment strategies, including chemotherapy, surgery, and radiotherapy, the 5-year overall survival rate in developed countries can achieve 90% [4,5]. However, WT is thought to be the most common solid tumor in sub-Saharan Africa and its outcome in resource-challenged settings continues to be sub-optimal, reaching a 25%–53% overall survival rate in low-income countries [6]. WT has the potential to grow into the inferior vena cava (and up to the right atrium) via extension into the renal vein. In particular, WT vascular extension to the inferior vena cava (IVC) occurs in 4–10% of cases and can reach the right atrium (RA) in 1%. In 50% of the cases, even when a cardiac extension is documented, clinical symptoms may be absent [7,8,9,10,11]. Reported risks of atrial extension are tricuspid valve (TV) obstruction and pulmonary embolism, necessitating aggressive surgical management of these tumors [12]. Data on WT with intracardiac extension, clinical presentation, and outcome in developing countries are limited.

The aim of the present study is to describe the prevalence of intracardiac extension in a consecutive population of WT patients enrolled in a large non-profit Ugandan hospital.

## 2. Methods

Patients were enrolled in a large non-profit Ugandan Hospital (Lacor Hospital, Gulu, North Uganda) for a 6-month period (September 2014–March 2015). Only patients with a histological diagnosis of WT were included.

### Echocardiography

In all patients, an echocardiogram was performed at admission (manufacturer: Esaote) by trained physicians at the Radiology Department of St. Mary’s Gulu Lachor Hospital. Complete standard 2D transthoracic analysis was accomplished. Left chambers’ volumes and left ventricle ejection fraction were measured by applying the biplane Simpson’s method [13]. Intracardiac extension of the tumors was measured by assessing the largest diameter of the mass and the orthogonal one.

The study was conducted according to the guidelines of the Declaration of Helsinki and approved by the Lacor Hospital Institutional Research Ethics Committee, in Gulu, Uganda (LHIREC 0212/03/2022). The need for informed consent was waived for this study, due to its nature as a retrospective, observational case series.

## 3. Results and Case Presentation

A total number of 25 patients with suspected WT were screened and 9 of them were excluded because the biopsy did not show a WT. Thus, 16 patients (male 50%; age 3 [2–4.75] years) were included in the final analysis. Table 1 reports the clinical and echocardiographic data of the population.

### 3.1. Case 1

Patient n°2 from Table 1, a 3 y/o child, presented with a 3-week history of abdominal distension, breathing difficulties, and swelling of the lower limbs. The cardiovascular system exam showed rhythmic heart sounds, a heart rate of 110 beats per minute, and a pansystolic murmur on the tricuspid area; the abdomen was grossly distended with a palpable mass in the right flank, hepatomegaly 12 cm below the costal margin, and splenomegaly. An abdomen ultrasound (US) showed an intra-abdominal tumor, involving the right kidney and the liver and extended to the IVC. An US-guided biopsy was performed, and histology confirmed the diagnosis of WT with triphasic histology (namely blastemal, epithelial, and stromal) with diffuse anaplasia. Cardiac echo showed a huge, mobile, cardiac mass (5.7 × 2.3 cm) attached to the right side of the interatrial septum (Figure 1), involving the tricuspid valve annulus, causing a “functional” tricuspid stenosis (Figure 2). The patient died of cardiogenic shock 7 days after admission before he could start chemotherapy and surgical treatment.

### 3.2. Case 2

Patient n°3 from Table 1, a 3 y/o child, presented with a 4-month history of breathing difficulties and abdominal distension. The cardiac examination was unremarkable. An US-guided biopsy confirmed the WT diagnosis and the patient was admitted to the pediatric ward for iv chemotherapy and further management. The cardiac echo showed a regular margin round mass (3.0 × 2.5 cm) in the RA (Figure 3). The patient was treated with iv chemotherapy (doxorubicin and vincristine) and, 4 weeks later, nephrectomy. Intraoperative examination showed a complete resolution of the IVC extension. Therefore, a thrombectomy was not performed. The patient was discharged 3 weeks after surgery. A pre-discharge echo confirmed a complete regression of the intracardiac mass. Due to logistical issues (distance from home to hospital, inability to travel, family economic problems), he never returned for a post-discharge follow-up visit.

### 3.3. Case 3

Patient n°16 from Table 1, a 2 y/o child, was admitted with a 1-month history of inappetence and body swelling. A 4 × 6 cm mass involving the left kidney suggestive of a tumor was noted at the abdomen US. Histology confirmed the diagnosis of WT, and a cardiac US showed the presence of a 2.0 × 1.5 cm mass in the RA with a fibrous pedicle attached to the IVC. The patient was treated with neoadjuvant anthracycline-based chemotherapy (doxorubicin), and abdominal surgery was successfully performed. At her 7-year follow-up, she presented no signs of abdomen or cardiac recurrence.

Thirteen more patients with suspected WT were screened with cardiac echo, showing absence of intracardiac tumor thrombus.

## 4. Discussion

In this case series, 16 consecutive pediatric patients with a histologically confirmed diagnosis of WT were admitted to a large non-profit hospital in North Uganda for a 6-month period. We documented a huge prevalence of intracardiac extension of the tumor (19%). As previously shown in a large cohort by Shamberger et al. [14], intravascular extension is not a rare finding in WT, involving 6% of cases, while intracardiac extension (RA) was reported in just 1% of cases. In this study, we found an about 20 times higher rate of RA involvement. Large studies confirmed a favorable outcome of children with WT and intravascular extension. In the study by Shamberger et al., the authors showed no significant difference in the survival rates of patients with intravascular extension in comparison with their counterparts without intravascular extension [14]. Notably, the 3-year survival rate was 90.0% for favorable histology tumors and 4.17% for unfavorable tumors, underlining how intracardiac extension is not a marker of poor prognosis per se, being mainly driven by the tumor histology. However, exporting these data in relation to a specific setting, such as a developing country, has significant limitations. In this setting, patients usually come to hospitals at an advanced stage [15]. In particular, transthoracic echocardiography is an examination that requires dedicated staff with specific skills as well as the availability of appropriate equipment (echocardiograph, probes, and software). These resources are not universally present at all centers in developing countries, reducing the accessibility of this method, which is considered a first-level investigation in developed countries. Even if the appropriate equipment is available, the possibility of having the examination performed by less experienced personnel (greater availability and lower costs) presents limitations which should be explored in greater depth in dedicated studies. More importantly, the constant availability of drugs for appropriate chemotherapy protocols is a major determinant of a WT patients’ outcome. In particular, as highlighted in case n°2, a neoadjuvant therapy can reduce (or even completely resolve) IVC and/or RA involvement. In these cases, it is therefore possible to reduce the need to associate an intravascular or even intracardiac surgery, which is often unavailable in a low-income setting.

Currently, most high-income countries reported that the 5-year survival rate is above 90% for children with localized disease [16,17]. However, the outcome in developing countries is still poor (the 5-year survival rate is less than 50%) [18]. Delayed diagnosis is one of the contributing factors for the low survival rate in these countries. It is caused by a lack of facilities for diagnosis and treatment, a lack of multidisciplinary collaboration, a lack of oncology referral centers and trained personnel, long distances to treatment centers, and also poor parental awareness on the early signs and symptoms of childhood malignancy [19]. Public health strategies aimed at increasing accessibility to diagnostic methods (echocardiography and computed tomography) in developing countries could lead to earlier diagnosis and thus improved prognoses in patients with WT. However, they appear limited by currently available resources.

Although no definitive conclusions can be drawn given the small sample size analyzed, the surgical and perioperative approach of these patients, already difficult in general, may become prohibitive in the presence of intracardiac extension. Additionally, in the study by Shamberger et al. [14], which enrolled patients from the American National Wilms Tumor Study Group, intravascular extension was associated with an increased frequency in surgical complications with an odds ratio of 2.2 by multivariate analysis [14]. In addition to the peculiar issue of the surgical event, other factors must also be considered as potentially limiting towards prognoses in pediatric patients with WT in a developing country. A lack of chemotherapeutics, possible perioperative infections, and limited availability of intensive care unit beds in case of serious complications after surgery are factors that may affect the prognoses of these patients. In addition to the more clinical aspects, socioeconomic issues can play a key role. In our case series, for example, a patient with WT intracardiac extension, patient n°2, who underwent successful surgery and was treated with chemotherapy, was then lost at follow-up due to logistic difficulties mainly related to his family’s financial problems, usually a less important issue in developed countries.

On the other side, our study demonstrated how treating WT with intracardiac extension is possible even in a developing country in sub-Saharan Africa, as shown by the favorable outcome of our Case 3 patient. A favorable outcome (irrespective of intracardiac extension) in WT patients treated in a developing country was also previously described by Bahoush et al. [20] in a retrospective study including 52 subjects (24 males and 28 females with an average age of 40 months). In this paper, the authors showed that female gender may be an independent negative prognostic factor for children with WT. 

Finally, WT management can be particularly challenging when it involves mechanical complications related to intracardiac invasion. In our Case 1 patient, a 3 y/o child with a large tumor and an extensive intracardiac extension, the cardiac mass was almost completely occluding the TV annulus, leading to a severe TV stenosis and, finally, to a cardiogenic shock. Kajal et al. [21] also reported a case of a WT with intracardiac extension, causing dynamic TV obstruction and prompt surgery with cardiopulmonary bypass, which is difficult to implement in a reality with limited resources (i.e., absence of cardiac surgeons). Indeed, the hospital where the study was conducted is not equipped with a cardiac surgery unit, which is missing in almost all health facilities in the state. Moreover, considering that perioperative treatment of WT is based on doxorubicin chemotherapy combinations (together with dactinomycin and vincristine), the systematic cardiac functional assessment of pediatric patients with WT before chemotherapy initiation, also in a low-income country context, is of paramount significance in order to adapt the chemotherapy schedule and dosage and to prevent potentially life-threating anthracycline-induced toxicities, particularly in patients with cardiac extension.

WT is thought to be the most common solid tumor in sub-Saharan Africa. The higher incidence would suggest different biology or environmental exposure between the African population and other regions where it is less common. Different genetic and/or environmental factors are believed to be responsible for this high prevalence. However, even if biological factors might influence incidence, prognosis, and outcome, the role of socioeconomic status is probably the most important prognostic factor.

### Limitations

This study has some limitations that must be discussed. First, the observation period (6 months) was limited to the period when dedicated staff were present on site to perform serial echocardiograms and collect data with a possible overestimation of the intracardiac extension incidence. On the other hand, this may indicate how not performing serial echocardiographic evaluations leads to underestimating the percentage of WT patients with IVC or RA involvement. In addition, the setting where the study was conducted was a large Ugandan non-profit hospital. In sub-Saharan Africa, health care can be highly variable from one region to another or sometimes even between same-region hospitals. Therefore, these results may not be applicable to all situations. Future studies are needed to confirm our data in larger populations and to assess the possible prognostic implications.

## 5. Conclusions

In our limited series, we found WT cardiac extension in three patients over 16 (19%). Cardiac echoes performed routinely can lead to a better staging, prognostic, and therapeutic assessment. In our setting, the intracardiac extension could be more frequent than previously reported and might have prognostic implications.

Apical 4 chambers view: a gross ovoid mass measuring 5.7 × 2.3 cm can be visualized in the RA. The mass engages the TV plane protruding into the right ventricle (Panel A). The subcostal view (Panel B) confirms the presence of a very mobile mass and shows a mild-moderate associated pericardial effusion (white arrow). The mass has an irregular and inhomogeneous structure with ultrasound images consistent with areas of vacuolization and/or tissue necrosis (white arrow) (Panel C).

The color Doppler image (Panel A) shows a significant acceleration of diastolic flow at the level of the TV plane, almost entirely occupied by the mass. Panel B shows the CW Doppler trace, which confirmed a severe TV stenosis. Abbreviations: TV: tricuspid valve; CW: continuous wave.

In the apical 4 chambers view, a 3.0 × 2.5 cm rounded mass is clearly seen inside the RA. Compared to the other case, the mass has limited motion with poor systolic-diastolic excursion (Panel A and B), staying away from the plane of the TV. Abbreviations: RA: right atrium; TV: tricuspid valve.

## Figures and Tables

**Figure 1 children-09-00743-f001:**
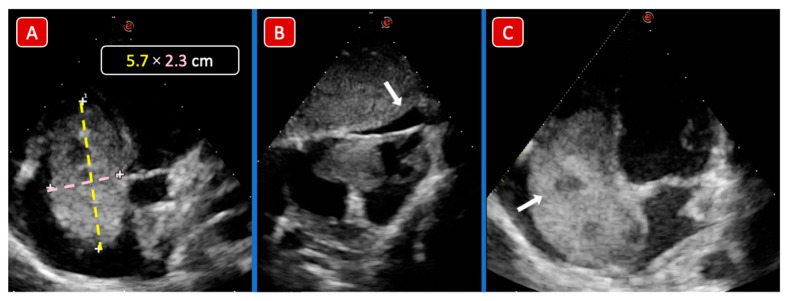
Case 1 echocardiographic images. Apical 4 chambers view: a gross ovoid mass measuring 5.7 × 2.3 cm can be visualized in the RA. The mass engages the TV plane protruding into the right ventricle (Panel (**A**)). Subcostal view (Panel (**B**)) confirms the pres-ence of a very mobile mass and shows a mild-moderate associated pericardial effusion (white arrow). The mass has an irregular and inhomogeneous structure with ultrasound images consistent with areas of vacuolization and/or tissue necrosis (white arrow) (Panel (**C**)).

**Figure 2 children-09-00743-f002:**
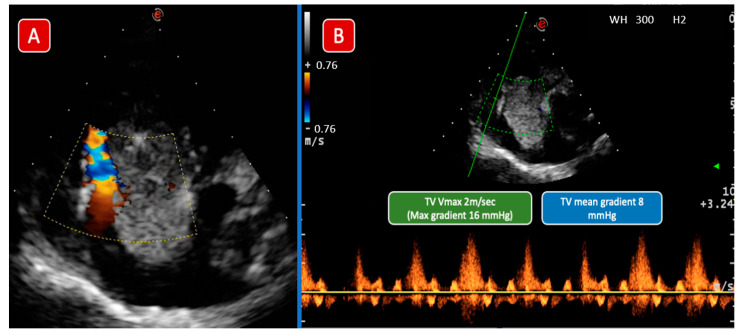
Case 1 echocardiographic images. Color Doppler image (Panel (**A**)) shows a significant acceleration of diastolic flow at the level of the TV plane, almost entirely occupied by the mass. Panel (**B**) shows the CW Doppler trace which confirmed a severe TV stenosis. Abbreviations: TV: Tricuspid Valve; CW: Continuous wave.

**Figure 3 children-09-00743-f003:**
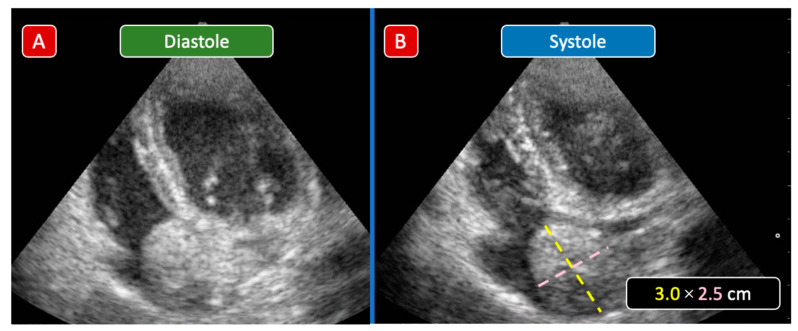
Case 2 echocardiographic images. Apical 4 chambers view: a 3.0 × 2.5 cm rounded mass is clearly seen inside the RA. Compared to the other case, the mass has limited motion with poor systolic-diastolic excursion (Panel (**A**,**B**)), remaining away from the plane of the TV. Abbreviations: RA: Right atrium; TV: Tricuspid Valve.

**Table 1 children-09-00743-t001:** Clinical and echocardiographic characteristics of 16 consecutive young patients with Wilms tumor (WT) admitted to the children’s ward. Abbreviations: LVEDV: left ventricle end diastolic volume; LVESV: left ventricle end systolic volume; EF: ejection fraction; TS: tricuspid stenosis; NA: not affected; NR: not reported. Average data (continuous variables) and prevalence data (dichotomous variables) are reported in the bottom line.

n	Age	Sex	Valves	LVEDV (mL)	LVESV (mL)	EF (%)	Intracardiac Extension	Dimension (cm)
1	7	F	NA	45	18	59	No	-
2	3	M	TS	NR	NR	55	Yes	5.7 × 2.3
3	3	M	NA	34	9	73	Yes	3 × 2.5
4	1	F	NA	17	7	60	No	-
5	4	F	NA	40	19	53	No	-
6	2	M	NA	22.7	9.2	60	No	-
7	1	F	NA	18.7	7.5	60	No	-
8	3	M	NA	24	10.8	55	No	-
9	3	M	NA	30	12.6	58	No	-
10	11	F	NA	55	18	68	No	-
11	8	F	NA	44	14	69	No	-
12	4	M	NA	45	15	66	No	-
13	7	M	NA	44	18	59	No	-
14	2	F	NA	26	8	62	No	-
15	3	M	NA	27.1	10.7	60	No	-
16	2	F	NA	NR	NR	60	Yes	2.0 × 1.5
	3 [2–4.75]	M (50%)	-	36.7 ± 11.4	14.0 ± 4.8	61.1 ± 5.4	3 (19%)	-

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
