# Peer review of "Unexpected Huge Prevalence of Intracardiac Extension of Wilms Tumor—A Single Center Experience from a Ugandan Hospital"

_children, 2022, doi:10.3390/children9050743_

Round 1
Reviewer 1 Report
- There are a number of typos and incorrect words used in this manuscript that should be corrected: For example: Shamberger is misspelled on page 4 line 120; page 3 line 93 “appetence” is not a word. Review of the entire manuscript for such errors should be undertaken.
- There is no reference to a research ethics board approval (or waiver if relevant) for this study – a statement should be provided that this was either exempt or at a minimum approved at the treating hospital by an ethics board.
- Convention in terms of confidentiality would typically not incorporate patient initials in a manuscript.
- It is somewhat surprising to me to see that all authors but one appear to be with Italian addresses. A better understanding of the role of the authors in the conduct of the echocardiograms in this patient setting would be helpful. What was the access then to echocardiography and what is it currently in this setting?
- Related to this and important to the understanding the context of the recommendation for African settings, the availability of echocardiogram both in terms of machinery and personnel expertise in Ugandan hospitals for this indication should be described. What is the expertise that is needed to conduct such studies? Can it be taught to non-experts in resource limited settings?
- The case series is gathered over a short period of just 6 months in 2014-5. Given the small number of cases with WT and therefore wide confidence interval estimates of whether this finding is happenstance or a true observation – why is a longer period not examined? A larger consecutive case series would be much more helpful. This series was also collected almost a decade ago… it would be a much more robust study with contemporary description of access to echocardiograms and results of testing - outcomes of childhood cancer in Africa have improved in the last decade... not clear to me that this report describes current circumstances - including from a presentation standpoint and diagnostic tool access.
- As this is a case series of patients focused on intracardiac tumor, the reporting of 13 unaffected patients in table 1 seems unnecessary to the paper.
- A significant part of the discussion provided relates to outcomes and biology for children with WT in African countries. This is not really relevant to the topic at hand which is the advanced presentation that is speculated to be prevalent. More discussion about how to improve access to tumor imaging including echo or CT, management of such patients if it is discovered would be helpful.
- Patient descriptions: They all seem incomplete. It is unclear in case 1 if the child received any chemo or cardiac surgical intervention before death. It is of note that it appears that case 2 received neoadjuvant chemo but there was no description of response or management of the intra-atrial tumor. Case 3 there was no description of the intracardiac management/follow up of this tumor.
- There are a number of papers published describing both advanced presentation and poorer outcomes of childhood cancers in Uganda and East Africa. These should be examined to help inform this study.
- Reference 14 seems incomplete.
Author Response
We thank reviewer #1 for his nice comments about our work. We have revised the manuscript accordingly and we think it has improved. Please see the point-by-point answers below
Reviewer #1
- There are a number of typos and incorrect words used in this manuscript that should be corrected: For example: Shamberger is misspelled on page 4 line 120; page 3 line 93 “appetence” is not a word. Review of the entire manuscript for such errors should be undertaken.
Thank you for pointing out the typos. The entire manuscript has been reviewed.
- There is no reference to a research ethics board approval (or waiver if relevant) for this study – a statement should be provided that this was either exempt or at a minimum approved at the treating hospital by an ethics board.
Thank you. The following statement has been added to the Methods section:
“The study was conducted according to the guidelines of the Declaration of Helsinki and approved by the Lacor Hospital Institutional Research Ethics Committee, Gulu, Uganda (LHIREC 0212/03/2022). Informed consent need was waived for this study, due to its retrospective, observational case series nature.”
- Convention in terms of confidentiality would typically not incorporate patient initials in a manuscript.
Thank you for your comment. Initials were removed from Table 1 and we replaced them in the text and in the abstract by numbers.
- It is somewhat surprising to me to see that all authors but one appear to be with Italian addresses. A better understanding of the role of the authors in the conduct of the echocardiograms in this patient setting would be helpful. What was the access then to echocardiography and what is it currently in this setting?
A team of Italian doctors worked at St. Mary's Lachor Hospital, Gulu, Uganda between 2014 and 2015, supporting local staff in performing echocardiograms. One of them (V.C.) was a permanent worker of the hospital at that time. Moreover 2 more local authors (supervisors) have been added: Jackson Kansiime, Cyprian Opira. The hospital has a permanent radiology department with staff trained also in echocardiography.
- Related to this and important to the understanding the context of the recommendation for African settings, the availability of echocardiogram both in terms of machinery and personnel expertise in Ugandan hospitals for this indication should be described. What is the expertise that is needed to conduct such studies? Can it be taught to non-experts in resource limited settings?
Thank you for your important comment. We agree. Following your suggestion, we have added these sentences to the discussion:
“In particular, transthoracic echocardiography is an examination that requires dedicated staff with specific skills as well as the availability of appropriate equipment (echocardiograph, probes, software). These resources are not universally present in all centers in developing countries, reducing the accessibility of this method, which is considered a first level investigation in developed countries. Even if the appropriate equipment is available, the possibility of having the examination performed by less experienced personnel (greater availability, lower costs) presents limitations which should be explored in greater depth in dedicated studies.”
- The case series is gathered over a short period of just 6 months in 2014-5. Given the small number of cases with WT and therefore wide confidence interval estimates of whether this finding is happenstance or a true observation – why is a longer period not examined? A larger consecutive case series would be much more helpful. This series was also collected almost a decade ago… it would be a much more robust study with contemporary description of access to echocardiograms and results of testing - outcomes of childhood cancer in Africa have improved in the last decade... not clear to me that this report describes current circumstances - including from a presentation standpoint and diagnostic tool access.
We totally agree with your comment. Unfortunately, the data presented in this study were collected during the time a team of Italian doctors worked at St. Mary's Lachor Hospital, Gulu, Uganda. The data, never published before, was only recently analysed and discussed. Moreover, even if echocardiograms are still performed daily in the same department, a routine data collection of these patients has been discontinued. We hope to conduct more studies on this topic in the future.
- As this is a case series of patients focused on intracardiac tumor, the reporting of 13 unaffected patients in table 1 seems unnecessary to the paper.
We thank reviewer #2 for this comment. We included Table 1 to present the characteristics of all consecutive young patients with Wilms tumor admitted to the Children Ward during the 6-month observation period, in order to provide a general overview of the population before focusing on the single cases descriptions. Otherwise the few clinical and echocardiographic characteristics showed in the table would have been left out.
- A significant part of the discussion provided relates to outcomes and biology for children with WT in African countries. This is not really relevant to the topic at hand which is the advanced presentation that is speculated to be prevalent. More discussion about how to improve access to tumor imaging including echo or CT, management of such patients if it is discovered would be helpful.
Thank you. We agree that improved accessibility to diagnostic methods in developing countries is central. However, we believe that this is beyond the scope of the current case series. Following your suggestion, we added this sentence to the discussion to underly the importance of this topic:
“Public health strategies aimed at increasing accessibility to diagnostic methods (echocardiography and computed tomography) in developing countries could lead to earlier diagnosis and thus improved prognosis in patients with WT. However, they appear limited by currently available resources.”
- Patient descriptions: They all seem incomplete. It is unclear in case 1 if the child received any chemo or cardiac surgical intervention before death. It is of note that it appears that case 2 received neoadjuvant chemo but there was no description of response or management of the intra-atrial tumor. Case 3 there was no description of the intracardiac management/follow up of this tumor.
Thank you for your observation. Please note we added the requested details in the three Cases reported.
We have also added, in the discussion section, a comment on the lack of availability of cardiac surgery:
“Indeed, the hospital where the study was conducted is not equipped with a cardiac surgery unit, which is missing in almost all health facilities in the state.”
- There are a number of papers published describing both advanced presentation and poorer outcomes of childhood cancers in Uganda and East Africa. These should be examined to help inform this study.
Thank you for your comment. Also reviewer #2 suggested to implement this aspect. The paper has been extended accordingly.
- Reference 14 seems incomplete.
This citation refers to a text book, we reported it as suggested by Pubmed “Cite” function.

Reviewer 2 Report
Dear authors,
Thank you for sharing your valuable information.
The suggestions given in this document are intended to improve your work.
Abstract:
According to the journal’s instructions: “The abstract should be a total of about 200 words maximum. The abstract should be a single paragraph and should follow the style of structured abstracts, but without headings.” Please remove the headings of “Backround”, “Case Series”, “Conclusion” and shorten the summary.
Introduction:
Backround was written but it’s better to write as the introduction. Backround is scarce. Wilms tumor, the parametres effecting WT and other important topics associated with WT should be explained in this section so that the readers can understand the scope of their research.
Methods:
This section is not comprehensive. The readers should learn echocardiogram and 2D transthoracic analysis and Simpson’s method. The methods that the researchers used must be explained. All methods should be given as a separate title (could be in italic form).
There is no data analysis for the participants. Which methods were used to analyse? I could not see the prevelance table as well. After these information, the researchers should share the results of study in the results section. The results section must be written. After the result section, they can start the discussion section.
Discussion: The discussion section was adequately compared with the results of other studies. I found this section good. Finish with limitations and future directions.
It would also be better if you write the recommendations section as a separate title at the end of the article.
Author Response
Reviewer #2
We thank reviewer #2 for his nice comments about our work. We have revised the manuscript accordingly and we think it has improved. Please see the point-by-point answers below
Dear authors, Thank you for sharing your valuable information. The suggestions given in this document are intended to improve your work.
We thank the reviewer #1 for his positive comment to our work.
Abstract: According to the journal’s instructions: “The abstract should be a total of about 200 words maximum. The abstract should be a single paragraph and should follow the style of structured abstracts, but without headings.” Please remove the headings of “Backround”, “Case Series”, “Conclusion” and shorten the summary.
Thank you. We have shortened the summary and removed the headings.
Introduction: Background was written but it’s better to write as the introduction. Backround is scarce. Wilms tumor, the parametres effecting WT and other important topics associated with WT should be explained in this section so that the readers can understand the scope of their research.
Thank you. As suggested we have changed the title from “Background” to “Introduction”. Moreover, we have expanded the introduction adding the following sentences (with the respective references)
“Multimodal strategies have led to a dramatic improvement in these patients’ prognosis. In fact, with optimized use of current treatment strategies, including chemotherapy, surgery, and radiotherapy, the 5-year overall survival in developed countries can achieve 90%. However, WT is thought to be the most common solid tumor in sub-Saharan Africa and its outcome in resource-challenged settings continues to be sub-optimal, reaching 25%–53% overall survival in low-income countries”
Methods: This section is not comprehensive. The readers should learn echocardiogram and 2D transthoracic analysis and Simpson’s method. The methods that the researchers used must be explained. All methods should be given as a separate title (could be in italic form).
Thank you for your comment. According to your suggestions the following paragraph has been created:
“Echocardiography
In all patients, an echocardiogram was performed at admission (Manufacturer Esaote) by trained physicians at the Radiology Department of St. Mary’s Gulu Lachor Hospital. Complete standard 2D transthoracic analysis was accomplished. Left chambers’ volumes and left ventricle ejection fraction were measured by applying the biplane Simpson’s method[10]. Intracardiac extension of the tumors was measured assessing the largest diameter of the mass and the orthogonal one.”
There is no data analysis for the participants. Which methods were used to analyse? I could not see the prevelance table as well. After these information, the researchers should share the results of study in the results section. The results section must be written. After the result section, they can start the discussion section.
Thank you for the comment. We reported in the Table 1 (see bottom line) average data (continuous variables) and prevalence data (dichotomous variables). The Table legend has been modified accordingly.
Moreover, following your suggestion we have added a new title for the section number 3, now re-named “Results and cases presentation” which includes now both the general overview of the population and the single cases descriptions.
Discussion: The discussion section was adequately compared with the results of other studies. I found this section good. Finish with limitations and future directions. It would also be better if you write the recommendations section as a separate title at the end of the article.
Thank you for your comments. As suggested we added a final paragraph with limitation and future directions
“Limitations
This study has some limitations that must be discussed. First, the observation period (6 months) was limited. In addition, the setting where the study has been conducted is a large Ugandan nonprofit hospital. In sub-Saharan Africa, health care can be highly variable from one region to another or sometimes even between same-region hospitals. Therefore, these results may not be applicable to all situations. Future studies are needed to confirm our data in larger populations and to assess the possible prognostic implications.”

Round 2
Reviewer 1 Report
The authors have adequately addressed the majority of comments although the fundamental flaw of a small case series collected remotely remains an issue.
Author Response
RESPONSE TO REVIEWERS
We thank the Editor for his comments about our work. We have revised the manuscript accordingly and we think it has improved. Please see the point-by-point answers below
This manuscript documents a series of Wilms tumor patients with an unusually high incidence of intracardiac tumor thrombus at a hospital in Uganda. The manuscript may be helpful to emphasize how resource limitations (late presentation due to lack of access to care, lack of full access to conventional neoadjuvant chemotherapy, lack of cardiac surgical or complex surgical oncology expertise) relate to poor outcomes in low-income countries.
Thank you for your comment. This was exactly the main message for our manuscript.
However, the case details are severely lacking, specifically regarding conventional data points for Wilms tumor (histology, presence of metastases, disease stage, etc, operative details). This must be improved before this manuscript can be published.
The patients were enrolled in a Ugandan nonprofit hospital where, at that time, digital storage of data (histology reports, operating minutes, etc.) was not constant. Therefore, some data were lost. However, we have fully revised the paper by adding the available details. In addition, main purpose of the paper was to describe the higher prevalence (and characteristics) of cardiac involvement in Wilms' tumors in a low-income setting
Line 29/abstract Correction: 16 patients with a histological diagnosis of 29 WT were screened in a 6-month period.
Thank you. Done as suggested
Line 56 – WT tends to invade vascular structures. I would rephrase this. WT has the potential to grow into the inferior vena cava (and up to the right atrium) via extension into the renal vein, but it does not invade other blood vessels. In fact, it is known for displacing blood vessels rather than encasing them (in the way neuroblastoma often encases blood vessels).
We agree with your comment. The sentence has been corrected accordingly.
Line 85 – specify the abbreviation bpm (beats per minute)
Corrected (we made the same correction also in the abstract)
Line 18 – Please change the title to “Results and Case Presentations”
Corrected
Line 88 – Please state more specific histologic findings from the pathology report (favorable histology, diffuse anaplasia, focal anaplasia, triphasic histology, etc) rather than “a picture consistent with WT” Please make sure that every abbreviation in the manuscript (RA, IVC, etc) is defined.
We agree with your comment and we have reported the new data coming from the histology report in the revised paper. “… histology confirmed the diagnosis of WT with triphasic histology (namely blastemal, epithelial and stromal) with diffuse anaplasia…”
Line 98 – Was only doxorubicin given or were vincristine and actinomycin-D both also given? For this case, was any IVC thrombectomy or atrial thrombectomy performed? How was the intravascular thrombus managed during the nephrectomy procedure? How long after chemotherapy started was the procedure performed?
The patient was treated with doxorubicine and vincristine. This and the other details available are now reported in the revised text.
“… Patient was treated with iv chemotherapy (doxorubicin and vincristine) and, 4 weeks later, nephrectomy. Intraoperative examination showed a complete resolution of the IVC extension. Therefore, thrombectomy was not performed. The patient has been discharged 3 weeks after surgery. A pre-discharge echo confirmed a complete regression of the intracardiac mass…”
It is confusing that Case 1 is patient number 2, Case 2 is patient number 3, and Case 3 is patient number 1. Please fix this.
Thank you. This is due to the fact that Table 1 reports the cases in chronological order. We have put the number reported in Table 1 in brackets close to the cases titles in the “Results and case presentation” section. For example: “Case 1 (Patient n°2, Table 1)”.
Please provide additional histologic details (as mentioned above) and additional surgical details (when performed) for each of the cases in the manuscript. The current descriptions are insufficient. It is not sufficient to say “abdominal surgery was performed” for a child with a renal tumor extending into the heart! We need the details!
The manuscript has been updated with all the additional data available. See above.
Line 110 – I would change the phrase “a normal heart picture” to “absence of intracardiac tumor thrombus”
Corrected. Thank you.
Line 114 – change to children’s ward; please use the term “non-profit”
The sentence has been rephrased according to your comment.
Line 163 – Here it says that the treatment of WT is doxorubicin, dactinomycin, and vincristine. In the above case reports doxorubicin is only mentioned. If only doxorubicin was given, was this because of resource limitations? This should be detailed and emphasized in the discussion section. The outcome of Wilms tumor patients with extensive intravascular thrombus (and especially cardiac extension) is dependent on adequate chemotherapy resources because many tumors can regress out of the heart during neoadjuvant therapy. The operation could then be performed without cardiac surgical expertise or cardiopulmonary bypass. These strategies may not be possible in the setting described in this series, but this logic should be emphasized to illustrate that people are not having good outcomes because of delayed presentation and severe resource limitations.
Thank you. This is also a very important comment. In our setting Doxorubicin was available while other chemotherapy (essentially vincristine) drugs were not always available (e.g., only for limited time and quantity). Given, as you mentioned, the importance of the neoadjuvant therapy in thrombosis resolution (see also our case 2), this is a crucial point. This is now stated in the discussion.
“…More importantly, the constant availability of drugs for appropriate chemotherapy protocols is a major determinant of WT patients’ outcome. In particular, as highlighted in the case n°2, a neoadjuvant therapy can produce a reduction (or even complete resolution) of IVC and/or RA involvement. In these cases, it is therefore possible to reduce the need to associate an intravascular or even intracardiac surgery, which is often unavailable in a low-income setting. …”
Table 1 – Make sure the cases match up with the patient numbers. Change it so they are consistent. I suggest for intracardiac extension stating yes or no rather than 0 or 1 in this table. The nn abbreviation is not defined in this table. Use the term Children’s Ward instead of Children ward here and throughout the manuscript. Define the +/- in this table are they ranges, standard deviations? Please replace commas with decimal points for example use 22.7 rather than 22,7.
Thank you. Table 1 has been modified according to your comments. We changed “nn” with “NA” (Not Affected) or “NR” (Not reported). Abbreviations have been added in the caption.
Line 173 – Please mention more clearly in the limitations section that the incidence/percentage of cardiac extension is likely overestimated due to the short observation period. Please state why the observation period was such a short period of time?
We agree with your comment and we have corrected the sentence as follows:
“First, the observation period (6 months) was limited to the period when dedicated staff were present on site to perform serial echocardiograms and collect data with a possible overestimation of the intracardiac extension incidence. On the other hand, this may indicate how not performing serial echocardiographic evaluations may underestimate the per-centage of WT patients with IVC or RA involvement.”

Reviewer 2 Report
Dear Authors,
Thank you for your revisons.
Author Response
thank you for your comments